# Sparse Parameter Adaptation for Fair Model Transfer Across Domains

**Sina Baharlouei**
eBay Search Ranking and Monetization
San Jose, CA, USA
sbaharlouei@ebay.com

**Minoo Ahmadi**
Department of Industrial & Systems Engineering
University of Southern California
Los Angeles, CA, USA
minooahm@usc.edu

## Abstract

Machine learning models trained in one domain often face significant challenges when deployed in a different domain due to distribution shifts, which can degrade both predictive performance and fairness. This paper studies the problem of transferring fair models from a source domain to a target domain where labeled data are scarce or unavailable, and only limited unlabeled data are accessible. We focus on scenarios where the original training data are inaccessible due to privacy or regulatory constraints, and fairness requirements must still be maintained in the target domain. To address these challenges, we propose a framework that regularizes model updates with sparsity-promoting penalties to adapt only a subset of parameters, enabling interpretable and reliable transfer. For linear models, we use an $\ell_1$-norm proximity term coupled with covariance-based fairness constraints, while for deep neural networks, we extend this idea via group sparse regularization. Additionally, we explore nonlinear fairness notions by incorporating $\chi^2$-divergence-based measures inspired by the FERMI [Lowy et al., 2022] framework. Empirical evaluations on the New Adult dataset demonstrate the effectiveness of our approach in transferring fair models from the source to target domain (different states) under limited target supervision. Our method achieves improved fairness-accuracy trade-offs while preserving interpretability, making it suitable for practical deployment in sensitive decision-making contexts such as credit eligibility across jurisdictions.

## 1 Introduction

Machine learning models are increasingly deployed in critical applications ranging from healthcare [Ahmad et al., 2018] and image processing [Krizhevsky et al., 2017] to education [Boselli et al., 2018] and cybersecurity [Xin et al., 2018]. While these models offer powerful tools for tackling complex societal problems, their uncritical deployment can lead to serious shortcomings such as biased predictions against minority groups [Angwin et al., 2016, Buolamwini and Gebru, 2018], susceptibility to adversarial attacks [Madry et al., 2017, Carlini and Wagner, 2017, Baharlouei et al., 2023b], and poor generalization to unseen settings [Arjovsky et al., 2019]. Therefore, ensuring that deployed models are trustworthy, fair, and compliant with global equality norms is of paramount importance [Act, 1964, Elford, 2023].

39th Conference on Neural Information Processing Systems (NeurIPS 2025) Workshop: RELIABLE ML FROM UNRELIABLE DATA.

Real-world examples highlight the societal harms of biased machine learning models. The COMPAS algorithm, studied by ProPublica [Angwin et al., 2016], demonstrated systemic bias against Black individuals in risk assessments. Similarly, facial recognition systems exhibit unequal performance, with significantly lower accuracy for darker-skinned females compared to lighter-skinned males [Buolamwini and Gebru, 2018]. These failures have motivated the development of fairness-aware learning algorithms, commonly categorized as pre-processing, in-processing, and post-processing techniques.

Pre-processing methods aim to transform the input data to reduce dependency on sensitive attributes prior to training [Kamiran and Calders, 2012, Zemel et al., 2013, Ustun et al., 2019]. Post-processing approaches adjust model predictions after training to achieve fairness criteria [Hardt et al., 2016, Alghamdi et al., 2022]. In contrast, in-processing methods incorporate fairness directly into the training objective, often by introducing fairness constraints or regularizers. For example,Zafar et al. [2017] minimize the covariance between sensitive attributes and predictions, while others utilize non-linear measures such as Rényi correlation[Baharlouei et al., 2020], $\chi^2$ divergence [Lowy et al., 2022], or Maximum Mean Discrepancy (MMD)[Prost et al., 2019]. In-processing methods may be model-specific[Wan et al., 2021, Aghaei et al., 2019] or generalizable across learning paradigms [Baharlouei et al., 2020, Lowy et al., 2022].

In-processing methods alternatively add fairness constraints or regularizers, penalizing dependence between sensitive attributes and output variables. [Zafar et al., 2017] utilizes covariance as the measure of independence between the sensitive attributes and the predictions. While such a measure is amenable to stochastic updates, it fails to capture correlations beyond linear. Alternatively, several non-linear measures such as Rényi correlation [Baharlouei et al., 2020], $\chi^2$ divergence [Lowy et al., 2022], $L_\infty$ distance [Donini et al., 2018], and Maximum Mean Discrepancy (MMD) [Prost et al., 2019] are proposed in the literature to establish the independence of the predictors and sensitive attributes. In-processing techniques can be model-specific [Wan et al., 2021, Aghaei et al., 2019] or generalizable to different training algorithms [Baharlouei et al., 2020, Lowy et al., 2022].

A common assumption in many fairness-aware methods is that the source and target data distributions are identical. However, real-world deployments often violate this assumption. For instance,Schrouff et al. [2022] show that a fairness-aware model trained on U.S. patients for predicting skin and hair conditions performs unfairly across age groups when applied to Colombian patients. Similarly,Ding et al. [2021] introduce the "New Adult" dataset, revealing significant distribution shifts across time and location, which adversely affect both performance and fairness of trained classifiers. These observations underscore the need for fairness-aware models that are robust to distributional shifts.

As machine learning systems are deployed in diverse geographic and demographic contexts, it becomes increasingly important to develop models that are both fair and robust to distributional changes. In this paper, we systematically characterize various scenarios of fair classification under distribution shift (see Table 1), based on the availability of labeled data, sensitive attributes, and access to source models or datasets. This taxonomy helps to clarify the assumptions, challenges, and objectives across different problem settings. Notably, we identify two underexplored yet realistic scenarios, and propose new methods to address them.

**Fair Classification in the Presence of Distribution Shifts Characterization:** A wide range of methods are proposed for transferring fair classifiers to a target domain containing of distribution shifts compared to the source domain. However, fair classification in the presence of distribution shift is not uniquely defined and characterized in the literature. In this section, we first describe different scenarios (see Table 1) under the fair classification in the presence of distribution shifts, and then we develop methods for two scenarios in which, the literature does not offer reliable solutions.

Assume two domains, source and target, with respective distributions $\mathcal{P}$source and $\mathcal{P}$target, and corresponding datasets $\mathcal{D}$source and $\mathcal{D}$target. We have access to $n$ labeled samples from the source and $m \ll n$ unlabeled samples from the target. The goal is to learn model parameters $\boldsymbol{\theta}^*$ within a hypothesis class $\mathcal{H}$, minimizing both prediction loss and a fairness regularization term:

$$\boldsymbol{\theta}^* = \operatorname*{arg\,min}_{\boldsymbol{\theta} \in \mathcal{H}} \mathbb{E}_{(\mathbf{x},y) \sim \mathcal{P}_{\text{target}}} \ell(h_{\boldsymbol{\theta}}(\mathbf{x}), y) + \lambda \rho_{\text{fairness}}\Big( \mathbb{P}(h_{\boldsymbol{\theta}}(\mathbf{x}), \mathbf{s}, y) \Big), \tag{1}$$

where $\rho_{\text{fairness}}$ is a measure of fairness as a function of the joint distribution of the model prediction, sensitive attributes, and the label. Popular choices of such measure is covariance [Zafar et al., 2017], Rényi [Grari et al., 2020, Baharlouei et al., 2020], $\chi^2$ divergence [Lowy et al., 2022], and in a more general case, the family of $f$-divergences [Baharlouei et al., 2023a] between the classifier's

decision boundary and the sensitive attribute(s). In this paper, we consider both covariance (due to its prevalence) and $f$-divergences due to its generality.

| $\mathcal{D}_{\text{source}}$ Available | $S_{\text{target}}$ Available | $Y_{\text{target}}$ Available | Problem Name | Notable Studies |
|---|---|---|---|---|
| Yes | No | No | Distributionally Robust Optimization | [Taskesen et al., 2020, Baharlouei and Razaviyayn, 2023] |
| No | Yes | Yes | Fair Model Transfer | [Lu et al., 2023, Schrouff et al., 2022] |
| Yes | No | Yes | Fair Domain Adaptation | [Ding et al., 2021], **Our Method** |
| Yes | Yes | No | Classic Domain Adaptation | [Ganin et al., 2016, Long et al., 2015] |
| Yes | No | No | Unsupervised Fair Domain Adaptation | [Donini et al., 2018, Prost et al., 2019] |
| No | No | Yes | Target-Only Fair Learning | [Hardt et al., 2016, Zafar et al., 2017] |

Table 1: Different scenarios for fair domain adaptation based on data availability

We list four possible scenarios in which the fair classification in the presence of the distribution shift can be studied:

1. **Fair Classification with No Data from the Target Domain** If we do not have access to any data from the test domain, the most common idea is to formulate the task as a [distributionally] robust optimization problem [Taskesen et al., 2020, Rezaei et al., 2021, Baharlouei and Razaviyayn, 2023]. While these methods demonstrate better performance on the target domains containing distribution shifts, their restrictive assumptions on not having access to zero data from the target domain neglect many real scenarios where a limited number of samples from the target distribution is available.

2. **Fair Model Transfer**: In this scenario, a fair model is already learned on the training domain, and the goal is to transfer the model to the target domain where a limited set of data points are available ($\mathcal{D}_{\text{target}}$). We can categorize this scenario as pre-training and fine-tuning stages. Note that, the main challenge in this scenario is that we do not have access to the training (source) data anymore, and fine-tuning of the model should be exclusively performed on the limited target data. We investigate this scenario in Section 2.

3. **Limited access to Data with Unknown Sensitive Attribute from the Target Domain**: This realistic scenario is studied in [Lu et al., 2023], where due to the privacy concerns on the target domain, only a limited number of samples from the target domain with anonymous sensitive attribute is available. This paper also considers another scenario, where a small number of samples with known sensitive attribute is given from the target domain.

4. **Limited Access to Unlabeled Data from the Target Distribution** This is the conventional domain adaptation problem with fairness-aware constraints. Such a scenario can happen due to the privacy concerns as well. Surprisingly, this scenario is not explored in the context of algorithmic fairness.

In summary, this paper makes the following contributions: (1) we provide a unified taxonomy of fair classification under distributional shifts, clarifying problem settings based on data availability; (2) we identify two underexplored scenarios: fair model transfer and unsupervised fair domain adaptation, and highlight their practical importance; and (3) we develop first-order algorithms that address fairness under these settings, with convergence guarantees and empirical validation on real datasets.

## 2   Transferring Fair Models Across Domains

Consider a scenario in which a fair model has already been trained on a source domain, but the original training data is unavailable due to privacy constraints. For instance, a financial institution may develop a credit card eligibility model using a large, labeled dataset of U.S. customers. The institution now seeks to deploy this model in Canada, where demographic profiles, credit regulations, and economic conditions differ significantly. However, no labeled data are available in the target domain (Canada), and only limited unlabeled customer data can be collected due to regulatory and privacy concerns. This shift across domains raises two key challenges: (1) the model, trained on U.S. data, may perform poorly or unfairly in the Canadian context; and (2) any modifications to the model must remain interpretable to satisfy legal and organizational transparency requirements. Interpretability is critical for ensuring regulatory compliance and for maintaining trust in automated decision-making systems deployed across jurisdictions [Yang et al., 2025].

Let $\boldsymbol{\theta}^*_{\text{source}}$ denote the optimal parameters of the fair model trained on the source domain:

$$\boldsymbol{\theta}^*_{\text{source}} = \arg\min_{\boldsymbol{\theta} \in \mathcal{H}} \mathbb{E}_{(\mathbf{x},y) \sim \mathcal{P}_{\text{source}}} \Big[ \ell\big(h_{\boldsymbol{\theta}}(\mathbf{x}), y\big) \Big] + \rho_{\text{fairness}}\Big( \mathbb{P}_{\text{source}}(h_{\boldsymbol{\theta}}(\mathbf{x}), \mathbf{s}, y) \Big), \tag{2}$$

Our goal is to adapt this model to the target domain, such that the resulting classifier remains both predictive and fair. Since labeled target data are unavailable, we use a small set of unlabeled target-domain samples and regularize model adaptation using the pre-trained parameters. The fair model transfer objective becomes:

$$\min_{\boldsymbol{\theta}} \quad \mathbb{E}_{(\mathbf{x},y) \sim \hat{\mathcal{P}}_{\text{target}}} [\ell(h_{\boldsymbol{\theta}}(\mathbf{x}), y)] + \alpha \cdot \rho_{\text{fairness}} \Big( \hat{\mathbb{P}}_{\text{target}}(h_{\boldsymbol{\theta}}(\mathbf{x}), \mathbf{s}, y) \Big) + \lambda \|\boldsymbol{\theta} - \boldsymbol{\theta}^*_{\text{source}}\|^q_p \tag{3}$$

Here, $\hat{\mathbb{P}}_{\text{target}}$ denotes the empirical distribution of the target domain ($m$ given samples from the target data), $\ell(\cdot, \cdot)$ is the predictive loss (e.g., cross-entropy), $\rho_{\text{fairness}}$ encodes a group fairness notion (e.g., demographic parity or equalized odds as the notion) violation, and $\lambda \|\boldsymbol{\theta} - \boldsymbol{\theta}^*_{\text{source}}\|^q_p$ penalizes deviation from the source model to preserve learned structure.

We focus on the case of $\ell_1$ norm (i.e., $p = q = 1$) which induces sparse updates from the source model. This means that only a subset of the parameters are adjusted in the target domain, while the rest remain unchanged. This is especially beneficial under limited data, as it helps prevent overfitting and promotes interpretability. In the case of a linear model (e.g., logistic regression), sparse adaptation reveals which features behave differently across domains—enhancing transparency and auditability.

To illustrate this concretely, we analyze this formulation in the simplest case where the predictor model is a linear classifier $h_{\boldsymbol{\theta}}(\mathbf{x}) = \mathbf{x}^\top \boldsymbol{\theta}$, and the fairness loss is based on the covariance between the sensitive attribute $s$ and the decision boundary proposed by Zafar et al. [2017]:

$$\rho_{\text{fairness}}(\boldsymbol{\theta}) = \big|\text{Cov}(s, \mathbf{x}^\top \boldsymbol{\theta})\big| = \big|\mathbb{E}[s \cdot \mathbf{x}^\top \boldsymbol{\theta}] - \mathbb{E}[s] \cdot \mathbb{E}[\mathbf{x}^\top \boldsymbol{\theta}]\big|. \tag{4}$$

This fairness penalty is linear in $\boldsymbol{\theta}$, and the full objective becomes:

$$\min_{\boldsymbol{\theta}} \quad \frac{1}{m} \sum_{i=1}^{m} \ell\big(h_{\boldsymbol{\theta}}(\mathbf{x}_i), y_i\big) + \alpha \big|\boldsymbol{\mu}^\top \boldsymbol{\theta}\big| + \lambda \|\boldsymbol{\theta} - \boldsymbol{\theta}^*_{\text{source}}\|_1, \tag{5}$$

where $\ell(h_{\boldsymbol{\theta}}(\mathbf{x}_i), y_i) = -y_i \log h_{\boldsymbol{\theta}}(\mathbf{x}_i) - (1 - y_i) \log(1 - h_{\boldsymbol{\theta}}(\mathbf{x}_i))$ is the logistic loss, $\boldsymbol{\mu} := \mathbb{E}[(s - \bar{s})\mathbf{x}]$ is a constant that can be calculated before training as:

$$\boldsymbol{\mu} = \frac{1}{m} \sum_{i=1}^{m} (s_i - \bar{s})\mathbf{x}_i, \quad \bar{s} = \frac{1}{m} \sum_{i=1}^{m} s_i,$$

and $\alpha, \lambda > 0$ are hyperparameters controlling fairness and sparsity penalties. The $\ell_1$ term biases the solution toward sparse updates: Only features strongly associated with unfairness are adjusted, while others remain fixed.

**Remark.** Given that $\ell$ is the cross-entropy (logistic) loss, Formulation (5) is a non-smooth convex function with respect to $\boldsymbol{\theta}$.

When no label information is available in the target domain, the fair model transfer objective simplifies to:

$$\min_{\boldsymbol{\theta}} \quad \big|\boldsymbol{\mu}^\top \boldsymbol{\theta}\big| + \lambda \|\boldsymbol{\theta} - \boldsymbol{\theta}^*_{\text{source}}\|_1, \tag{6}$$

This formulation aims to adjust the model parameters to improve fairness in the target domain, measured via the covariance-based fairness criterion, while remaining close to the original model trained on the source domain.

To illustrate the behavior of this objective, we consider a toy example in which both $\boldsymbol{\mu}$ and $\boldsymbol{\theta}^*_{\text{source}}$ are sampled as 100-dimensional Gaussian vectors. Figure 1 shows the number of differing coordinates between the optimized $\boldsymbol{\theta}$ and the original $\boldsymbol{\theta}^*_{\text{source}}$ as a function of the regularization parameter $\lambda$. As expected, increasing $\lambda$ results in fewer parameter updates. In the limit, as $\lambda \to \infty$, the solution converges to $\boldsymbol{\theta} = \boldsymbol{\theta}^*_{\text{source}}$ indicating no adaptation. This sparsity pattern highlights the trade-off between fairness improvement and model stability during domain transfer.

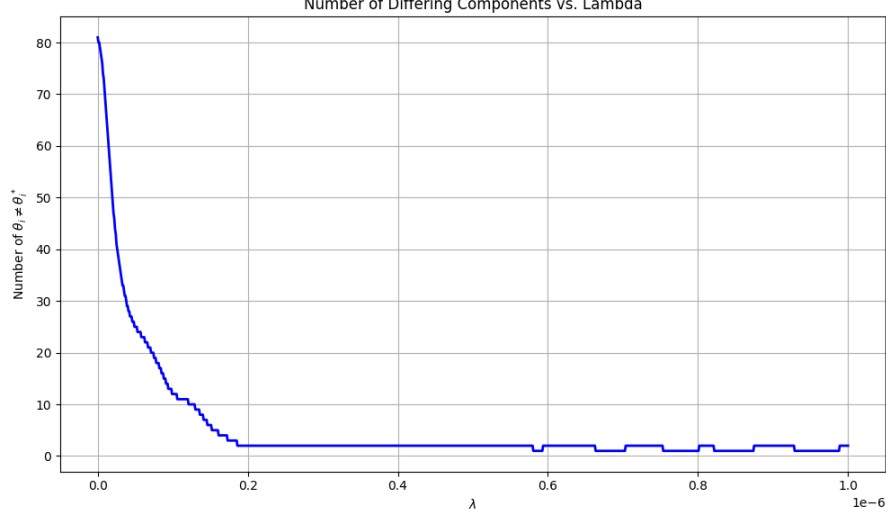

Figure 1: Number of components in $\boldsymbol{\theta}$ that differ from $\boldsymbol{\theta}^*$ as a function of $\lambda$. As $\lambda$ increases, the solution becomes more similar to $\boldsymbol{\theta}^*$ due to the increasing weight of the $\ell_1$ penalty.

## 2.1 Extension to Non-linear Notions of Fairness Violation

While covariance-based fairness measures (e.g., as in (4)) capture only linear dependencies between sensitive attributes and model predictions, real-world fairness violations often arise from complex, nonlinear relationships. To better address these, we extend our framework by incorporating the $\chi^2$-divergence via the exponential Rényi mutual information (ERMI) measure introduced in FERMI [Lowy et al., 2022].

Specifically, the ERMI is defined as:

$$D_R(\widehat{Y}; S) := \mathbb{E}\left\{\frac{p_{\widehat{Y},S}(h_{\boldsymbol{\theta}}(\mathbf{x}_i), \mathbf{s})}{p_{\widehat{Y}}(h_{\boldsymbol{\theta}}(\mathbf{x}_i))p_S(\mathbf{s})}\right\} - 1 = \sum_{j \in [m]} \sum_{r \in [k]} \frac{p_{\widehat{Y},S}(j,r)^2}{p_{\widehat{Y}}(j)p_S(r)} - 1, \qquad \text{(ERMI)}$$

where $\widehat{Y}$ is the discrete model prediction and $S$ is the sensitive attribute, with $m$ and $k$ denoting the number of possible values for $\widehat{Y}$ and $S$, respectively.

This measure captures nonlinear dependence between the model's output and sensitive attributes, encouraging statistical independence when minimized. Accordingly, the fair transfer objective becomes:

$$\min_{\boldsymbol{\theta}} \quad \frac{1}{m}\sum_{i=1}^m \ell\big(h_{\boldsymbol{\theta}}(\mathbf{x}_i), y_i\big) + \alpha \cdot D_R\left(h_{\boldsymbol{\theta}}(\mathbf{x}_i); \mathbf{s}\right) + \lambda \sum_{i=1}^d \|W_{:,i}^{(1)} - W_{:,i}^{*(1)}\|_2, \qquad (7)$$

In practice, ERMI can be efficiently estimated from discrete samples of $\mathbf{s}$ and the model's predictions.

## 2.2 Extension to Deep Neural Networks via Group Sparse Regularization

While the formulation in Eq. (5) enables interpretable transfer for linear models by sparsely modifying only those parameters associated with features affected by distributional shift, the same intuition does not directly apply to deep neural networks. In non-linear architectures, individual model weights are not directly associated with specific input features, making parameter-level sparsity harder to interpret.

To address this, we extend our approach using *group Lasso regularization* [Scardapane et al., 2016], which enables *feature-level interpretability* in neural networks. Specifically, we define groups of parameters based on the input-layer connections: for each input feature $i \in \{1, \ldots, d\}$, we define a group $G_i$ as the set of all weights connecting feature $i$ to neurons in the first hidden layer:

$$G_i = \left\{W_{:,i}^{(1)}\right\}, \qquad (8)$$

where $W_{:,i}^{(1)}$ denotes the $i$-th column of the first-layer weight matrix, i.e., all weights from feature $i$ to the first-layer neurons.

The fair model transfer objective for deep neural networks then becomes:

$$\min_{\boldsymbol{\theta}} \quad \frac{1}{m} \sum_{i=1}^{m} \ell\left(h_{\boldsymbol{\theta}}(\mathbf{x}_i), y_i\right) + \alpha \cdot D_R\left(h_{\boldsymbol{\theta}}(\mathbf{x}_i); \mathbf{s}\right) + \lambda \sum_{i=1}^{d} \left\| W_{:,i}^{(1)} - W_{:,i}^{*(1)} \right\|_2, \tag{9}$$

where $W_{:,i}^{*(1)}$ denotes the pre-trained weights from the source model. This *group-wise $\ell_2$ penalty* encourages *entire input features* to either remain unchanged or be updated as a whole. In particular, features that do not contribute to fairness violations under domain shift will retain their original weights (i.e., $\| W_{:,i}^{(1)} - W_{:,i}^{*(1)} \|_2 = 0$), while features needing adaptation will be adjusted collectively.

This formulation preserves the interpretability of the linear case by promoting *sparse feature-level changes*, even in the context of deep architectures. Thus, we can identify which features are responsible for fairness degradation in the target domain, supporting transparency and accountability in model deployment.

Algorithm 1 provides a proximal gradient approach for optimizing 9. At each iteration of the algorithm, we update the parameters in $\boldsymbol{\theta}$ that are not in the regularization term (smooth part) via gradient descent, while we apply an extra proximal operator to the weights in the regularization term. Notice that if the logistic regression loss is a special case of (9), where we have input and output (cross-entropy loss) in a given neural network.

---

**Algorithm 1** Proximal Gradient Algorithm for Fair Transfer with Group Lasso Regularization

---

**Require:** Initial weights $\boldsymbol{\theta}^{(0)}$, source weights $W^{*(1)}$, step size $\eta > 0$, regularization parameters $\alpha, \lambda$, max iterations $T$, data $\{(\mathbf{x}_i, y_i)\}_{i=1}^{m}$

**Ensure:** Updated weights $\boldsymbol{\theta}^{(T)}$

1: **for** $t = 0$ to $T - 1$ **do**
2:     Compute gradient of smooth loss and fairness terms:

$$\mathbf{g}^{(t)} = \nabla_{\boldsymbol{\theta}} \left( \frac{1}{m} \sum_{i=1}^{m} \ell(h_{\boldsymbol{\theta}^{(t)}}(\mathbf{x}_i), y_i) + \alpha \cdot D_R\left(h_{\boldsymbol{\theta}^{(t)}}(\mathbf{x}_i); \mathbf{s}\right) \right)$$

3:     Gradient descent step:
$$\boldsymbol{\theta}^{(t+1/2)} = \boldsymbol{\theta}^{(t)} - \eta \mathbf{g}^{(t)}$$

4:     **Proximal update for group Lasso regularization on first layer weights:**
5:     For each input feature group $i = 1, \ldots, d$:

$$W_{:,i}^{(1)(t+1)} = \text{prox}_{\eta\lambda\| \cdot - W_{:,i}^{*(1)}\|_2} \left( W_{:,i}^{(1)(t+1/2)} \right)$$

where the proximal operator for group Lasso with center $W_{:,i}^{*(1)}$ is:

$$\text{prox}_{\tau\| \cdot - W_{:,i}^{*(1)}\|_2}(v) = \begin{cases} W_{:,i}^{*(1)} + \left( 1 - \frac{\tau}{\|v - W_{:,i}^{*(1)}\|_2} \right)(v - W_{:,i}^{*(1)}), & \text{if } \|v - W_{:,i}^{*(1)}\|_2 > \tau \\ W_{:,i}^{*(1)}, & \text{otherwise} \end{cases}$$

6: **end for**
7: **return** $\boldsymbol{\theta}^{(T)}$

---

**Note:** A natural extension of our model transfer framework to the unsupervised domain adaptation setting is to remove the predictive loss term from (9), resulting in the following objective:

$$\min_{\boldsymbol{\theta}} \quad D_R\left(h_{\boldsymbol{\theta}}(\mathbf{x}); \mathbf{s}\right) + \lambda \sum_{i=1}^{d} \left\| W_{:,i}^{(1)} - W_{:,i}^{*(1)} \right\|_2, \tag{10}$$

This formulation adapts the model solely based on minimizing fairness violation in the target domain, using only unlabeled feature data. However, a potential limitation is that it does not explicitly preserve the predictive performance of the source model. Incorporating mechanisms that leverage $\mathbf{x}$ to maintain source-domain accuracy during adaptation remains an important direction for future work.

# 3 Results

To evaluate the effectiveness of our proposed fair model transfer framework, we conduct experiments on the *New Adult* dataset introduced by Ding et al. [2021]. This dataset extends the widely-used Adult dataset [Dua and Graff, 2017] by incorporating temporal and geographic diversity—spanning multiple U.S. states and years—thus capturing natural distribution shifts in demographic and socioeconomic features. These characteristics make it well-suited for studying domain adaptation under fairness constraints.

## 3.1 Experimental Setup

We consider the task of transferring a fair classifier trained on California data (source domain) to Texas data (target domain). These states differ significantly in demographics and economic conditions, resulting in shifts in feature distributions and potentially different relationships between features and outcomes such as credit eligibility. Crucially, we assume that label information is unavailable in the target domain due to privacy and regulatory constraints, reflecting a realistic scenario. Only the pre-trained model parameters from California and a small number of labeled or unlabeled Texas samples are available for adaptation. Our objective is to evaluate whether our sparse, interpretable adaptation method can maintain both predictive accuracy and fairness when transferring across these domains.

## 3.2 Baselines and Evaluation Protocol

We compare our method against the following baselines:

- [Lowy et al., 2022] trained on source-only and target-only data.
- [Zafar et al., 2017] trained on the target data (excluded from source-only due to consistently worse performance).
- Distributionally Robust FERMI [Baharlouei and Razaviyayn, 2023].
- Our method with both $\ell_1$ and $\ell_2$ proximity terms.

All models are initially trained on the source domain such that the demographic parity fairness violation is approximately 0.02. For adaptation, we tune the hyperparameters $\alpha(125.4)$ and $\lambda(3.5)$ to ensure the fairness violation on the adaptation set remains close to 0.02. We then evaluate accuracy and fairness on the full Texas dataset.

## 3.3 Logistic Regression Results: Performance Across Data Regimes

$\ell_1$ **regularization yields best trade-off with 100 labeled samples.** As shown in Table 2, when the number of labeled samples in the target domain is very limited (100), our adaptation method significantly outperforms the baselines in both accuracy and fairness. The $\ell_1$ regularizer yields the best result, likely due to its sparsity-inducing nature, which is advantageous in low-data regimes.

$\ell_2$ **regularization performs best with 1000 labeled samples.** Table 3 shows that when more labeled samples are available, adaptation still provides a significant boost. Interestingly, $\ell_2$ regularization performs slightly better than $\ell_1$, suggesting that smoother adaptation may be more effective when sufficient target data is available.

**Direct training optimal with 10,000+ labeled samples.** With large amounts of target data, Table 4 confirms that direct training on the target domain performs best. In this setting, adaptation is less necessary, and regularization offers no clear benefit.

Table 2: Performance comparison when training on California data and adapting with **100** labeled samples from Texas. Our $\ell_1$ method achieves 73.45% accuracy while maintaining fairness at 0.020, outperforming target-only training (62.44%) by 11 percentage points.

| Method | Source Accuracy (%) | Target Accuracy (%) | Target Fairness |
|---|---|---|---|
| [Lowy et al., 2022] on Source | 76.48% | 71.17% | 0.054 |
| [Lowy et al., 2022] on Target | 76.48% | 62.44% | 0.023 |
| Dr-FERMI | 73.31% | 70.59% | 0.034 |
| [Zafar et al., 2017] | - | 70.44% | 0.063 |
| Ours ($\ell_1$ norm) | – | **73.45%** | **0.020** |
| Ours ($\ell_2$ norm) | – | 72.11% | 0.020 |

Table 3: Performance comparison when training on California data and adapting with **1000** labeled samples from Texas. With moderate target data, $\ell_2$ regularization achieves 75.16% accuracy with 0.0205 fairness, showing smoother adaptation is effective when sufficient samples are available.

| Method | Source Accuracy | Target Accuracy | Target Fairness |
|---|---|---|---|
| [Lowy et al., 2022] on Target | 76.48% | 70.89% | 0.021 |
| [Zafar et al., 2017] | - | 70.44% | 0.0504 |
| Ours ($\ell_1$ norm) | - | 74.92% | 0.0204 |
| Ours ($\ell_2$ norm) | - | **75.16%** | **0.0205** |

### 3.4 Deep Neural Network Results: Group Sparsity Maintains Interpretability

Next, we consider a three-layer neural network with 32 and 128 hidden units respectively. Here, we use Algorithm 1 to optimize the objective function (9). Notably, Dr-FERMI does not provide a neural network implementation, so it is excluded from this comparison.

As shown in Table 5, our approach with $\ell_1$ regularization significantly outperforms other methods in target accuracy and fairness. The performance of $\ell_2$ regularization degrades substantially, which we attribute to the non-convexity and non-uniqueness of neural network training: small $\ell_2$ distances may still lead to dramatically different functional outcomes.

## 4 Conclusion

In this work, we presented a framework for transferring fair machine learning models across domains in a manner that is both interpretable and effective under data and fairness constraints. Our approach adapts a pre-trained fair model from a source domain to a target domain using sparse updates, which allows only a small subset of model parameters to change—promoting interpretability and robustness, especially in low-data regimes.

We introduced a general formulation that incorporates fairness constraints (including non-linear ones such as chi-squared divergence) alongside proximity regularization to the source model. We demonstrated that $\ell_1$-based regularization is particularly effective in sparse adaptation settings, while group sparsity extensions make our method suitable for deep neural networks. Additionally, we showed that our approach outperforms baselines on the New Adult dataset in scenarios with limited target domain data.

Empirical results highlighted that when large amounts of target data are available, training a new model directly on the target domain yields the best performance. However, in realistic settings where target labels are scarce, our adaptation method offers a compelling trade-off between fairness, accuracy, and interpretability.

Future work includes extending our method to fully unsupervised settings with mechanisms to preserve source-domain accuracy, evaluating temporal distribution shifts alongside geographic shifts, and validating on additional datasets to establish broader generalizability.

## References

Civil Rights Act. Civil rights act of 1964. *Title VII, Equal Employment Opportunities*, 1964.

Table 4: Performance comparison when training on California data and adapting with **10000** labeled samples from Texas. With abundant target data, direct training (75.81% accuracy) outperforms adaptation, indicating diminishing returns from regularization when labels are plentiful.

| Method | Source Accuracy | Target Accuracy | Target Fairness |
|---|---|---|---|
| [Lowy et al., 2022] on Target | 76.48% | **75.81%** | **0.0202** |
| [Zafar et al., 2017] | - | 70.44% | 0.041 |
| Ours ($\ell_1$ norm) | - | 74.92% | 0.0203 |
| Ours ($\ell_2$ norm) | - | 75.38% | 0.0202 |

Table 5: Performance comparison when training the neural network on California data and adapting with **5000** labeled samples from Texas. Group sparse $\ell_1$ regularization achieves 77.22% accuracy with 0.023 fairness, outperforming baselines while maintaining feature-level interpretability.

| Method | Source Accuracy | Target Accuracy | Target Fairness |
|---|---|---|---|
| [Lowy et al., 2022] | 79.73% | 74.96% | 0.026 |
| [Zafar et al., 2017] | - | 73.62% | 0.038 |
| Ours ($\ell_1$ norm) | - | **77.22%** | **0.023** |
| Ours ($\ell_2$ norm) | - | 71.42% | 0.025 |

Sina Aghaei, Mohammad Javad Azizi, and Phebe Vayanos. Learning optimal and fair decision trees for non-discriminative decision-making. In *Proceedings of the AAAI conference on artificial intelligence*, pages 1418–1426, 2019.

Muhammad Aurangzeb Ahmad, Carly Eckert, and Ankur Teredesai. Interpretable machine learning in healthcare. In *Proceedings of the 2018 ACM international conference on bioinformatics, computational biology, and health informatics*, pages 559–560, 2018.

Wael Alghamdi, Hsiang Hsu, Haewon Jeong, Hao Wang, Peter Michalak, Shahab Asoodeh, and Flavio Calmon. Beyond adult and compas: Fair multi-class prediction via information projection. *Advances in Neural Information Processing Systems*, 35:38747–38760, 2022.

Julia Angwin, Jeff Larson, Surya Mattu, and Lauren Kirchner. Machine bias. In *Ethics of Data and Analytics*, pages 254–264. Auerbach Publications, 2016.

Martin Arjovsky, Léon Bottou, Ishaan Gulrajani, and David Lopez-Paz. Invariant risk minimization. *arXiv preprint arXiv:1907.02893*, 2019.

Sina Baharlouei and Meisam Razaviyayn. Dr. fermi: A stochastic distributionally robust fair empirical risk minimization framework. *arXiv preprint arXiv:2309.11682*, 2023.

Sina Baharlouei, Maher Nouiehed, Ahmad Beirami, and Meisam Razaviyayn. Rényi fair inference. In *International Conference on Learning Representations*, 2020.

Sina Baharlouei, Shivam Patel, and Meisam Razaviyayn. f-ferm: A scalable framework for robust fair empirical risk minimization. *arXiv preprint arXiv:2312.03259*, 2023a.

Sina Baharlouei, Fatemeh Sheikholeslami, Meisam Razaviyayn, and Zico Kolter. Improving adversarial robustness via joint classification and multiple explicit detection classes. In Francisco Ruiz, Jennifer Dy, and Jan-Willem van de Meent, editors, *Proceedings of The 26th International Conference on Artificial Intelligence and Statistics*, volume 206 of *Proceedings of Machine Learning Research*, pages 11059–11078. PMLR, 25–27 Apr 2023b.

Roberto Boselli, Mirko Cesarini, Fabio Mercorio, and Mario Mezzanzanica. Classifying online job advertisements through machine learning. *Future Generation Computer Systems*, 86:319–328, 2018.

Joy Buolamwini and Timnit Gebru. Gender shades: Intersectional accuracy disparities in commercial gender classification. In *Conference on fairness, accountability and transparency*, pages 77–91. PMLR, 2018.

Nicholas Carlini and David Wagner. Towards evaluating the robustness of neural networks. In *2017 IEEE Symposium on Security and Privacy (SP)*, pages 39–57, 2017. doi: 10.1109/SP.2017.49.

Frances Ding, Moritz Hardt, John Miller, and Ludwig Schmidt. Retiring adult: New datasets for fair machine learning. *Advances in Neural Information Processing Systems*, 34:6478–6490, 2021.

Michele Donini, Luca Oneto, Shai Ben-David, John S Shawe-Taylor, and Massimiliano Pontil. Empirical risk minimization under fairness constraints. *Advances in neural information processing systems*, 31, 2018.

Dheeru Dua and Casey Graff. Uci machine learning repository, 2017. URL `http://archive.ics.uci.edu/ml`.

Gideon Elford. Equality of Opportunity. 2023.

Yaroslav Ganin, Evgeniya Ustinova, Hana Ajakan, Pascal Germain, Hugo Larochelle, François Laviolette, Mario March, and Victor Lempitsky. Domain-adversarial training of neural networks. *Journal of machine learning research*, 17(59):1–35, 2016.

Vincent Grari, Sylvain Lamprier, and Marcin Detyniecki. Fairness-aware neural rényi minimization for continuous features. In *Twenty-Ninth International Joint Conference on Artificial Intelligence and Seventeenth Pacific Rim International Conference on Artificial Intelligence {IJCAI-PRICAI-20}*, pages 2262–2268. International Joint Conferences on Artificial Intelligence Organization, 2020.

Moritz Hardt, Eric Price, and Nati Srebro. Equality of opportunity in supervised learning. *Advances in neural information processing systems*, 29, 2016.

Faisal Kamiran and Toon Calders. Data preprocessing techniques for classification without discrimination. *Knowledge and information systems*, 33(1):1–33, 2012.

Alex Krizhevsky, Ilya Sutskever, and Geoffrey E Hinton. Imagenet classification with deep convolutional neural networks. *Communications of the ACM*, 60(6):84–90, 2017.

Mingsheng Long, Yue Cao, Jianmin Wang, and Michael Jordan. Learning transferable features with deep adaptation networks. In *International conference on machine learning*, pages 97–105. PMLR, 2015.

Andrew Lowy, Sina Baharlouei, Rakesh Pavan, Meisam Razaviyayn, and Ahmad Beirami. A stochastic optimization framework for fair risk minimization. *tmlr*, 2022.

Yiwei Lu, Guojun Zhang, Sun Sun, Hongyu Guo, and Yaoliang Yu. $f$-micl: Understanding and generalizing infonce-based contrastive learning. *Transactions on Machine Learning Research*, 2023.

Aleksander Madry, Aleksandar Makelov, Ludwig Schmidt, Dimitris Tsipras, and Adrian Vladu. Towards deep learning models resistant to adversarial attacks. *arXiv preprint arXiv:1706.06083*, 2017.

Flavien Prost, Hai Qian, Qiuwen Chen, Ed H Chi, Jilin Chen, and Alex Beutel. Toward a better trade-off between performance and fairness with kernel-based distribution matching. *arXiv preprint arXiv:1910.11779*, 2019.

Ashkan Rezaei, Anqi Liu, Omid Memarrast, and Brian D Ziebart. Robust fairness under covariate shift. In *Proceedings of the AAAI Conference on Artificial Intelligence*, volume 35, pages 9419–9427, 2021.

Simone Scardapane, Danilo Comminiello, Amir Hussain, and Aurelio Uncini. Group sparse regularization for deep neural networks. *IEEE Transactions on Neural Networks and Learning Systems*, 28(7):1555–1565, 2016.

Jessica Schrouff, Natalie Harris, Sanmi Koyejo, Ibrahim M Alabdulmohsin, Eva Schnider, Krista Opsahl-Ong, Alexander Brown, Subhrajit Roy, Diana Mincu, Christina Chen, et al. Diagnosing failures of fairness transfer across distribution shift in real-world medical settings. *Advances in Neural Information Processing Systems*, 35:19304–19318, 2022.

Bahar Taskesen, Viet Anh Nguyen, Daniel Kuhn, and Jose Blanchet. A distributionally robust approach to fair classification, 2020.

Berk Ustun, Yang Liu, and David Parkes. Fairness without harm: Decoupled classifiers with preference guarantees. In *International Conference on Machine Learning*, pages 6373–6382. PMLR, 2019.

Mingyang Wan, Daochen Zha, Ninghao Liu, and Na Zou. Modeling techniques for machine learning fairness: A survey. *CoRR*, abs/2111.03015, 2021. URL `https://arxiv.org/abs/2111.03015`.

Yang Xin, Lingshuang Kong, Zhi Liu, Yuling Chen, Yanmiao Li, Hongliang Zhu, Mingcheng Gao, Haixia Hou, and Chunhua Wang. Machine learning and deep learning methods for cybersecurity. *Ieee access*, 6:35365–35381, 2018.

Shiqi Yang, Ziyi Huang, Wengran Xiao, and Xinyu Shen. Interpretable credit default prediction with ensemble learning and shap. *arXiv preprint arXiv:2505.20815*, 2025.

Muhammad Bilal Zafar, Isabel Valera, Manuel Gomez Rogriguez, and Krishna P Gummadi. Fairness constraints: Mechanisms for fair classification. In *Artificial intelligence and statistics*, pages 962–970. PMLR, 2017.

Rich Zemel, Yu Wu, Kevin Swersky, Toni Pitassi, and Cynthia Dwork. Learning fair representations. In *International conference on machine learning*, pages 325–333. PMLR, 2013.

