# OpenReview forum: "Sparse Parameter Adaptation for Fair Model Transfer Across Domains"
_NeurIPS.cc/2025/Workshop/Reliable_ML — NeurIPS 2025 - Reliable ML Workshop_

### Official Review · Reviewer_hmxa · 2025-09-19
**Well motivated problem. Significant results. Shaky experimental evaluation.**

**Rating:** 4
**Confidence:** 3

**Review:**

Summary:
The authors address the issue of transferring fair models from a source domain to a target domain where labeled data are scarce or nonexistent. They provide a unified taxonomy of fair classification under distributional shifts and identify two underexplored scenarios: fair model transfer and unsupervised fair domain adaptation. Finally, they develop first-order algorithms that address fairness in these settings with convergence guarantees and empirical validation on real datasets.
Strengths:
1) The authors provide a unified taxonomy of fair classification under distributional shifts. The taxonomy seems logical and useful.
2) The problem is well-motivated and within the scope of this workshop.
3) The example of transferring California data to Texas in the experimental section makes sense in the context of the paper.
Weaknesses:
1) The first-order algorithms presented are not particularly novel.
2) I am also concerned about the experimental results claimed by the authors. First, I believe that a single dataset with one experiment for each target sample is insufficient for comparing methods. Most importantly, the experimental setup is not clearly stated. What do the labels represent? What are the sensitive attributes? Was any procedure applied to outliers? Even if the authors consider these questions to be trivial, I believe the answers should always be stated for the experimental results to be reproducible.
Suggestions for the authors:
I believe that, with a more thorough and well-stated experimental evaluation, the ideas in this paper could be significant.

---

### Official Review · Reviewer_5FCh · 2025-09-20
**Strong motivation and organized related work, but clearer takeaways and temporal evaluation would make the paper stronger**

**Rating:** 9
**Confidence:** 4

**Review:**

# *Summary*

This paper works on the challenge of transferring compliant fair models from one domain to another where training data may not be available or are limited , and distribution shifts are expected. Empirical results showed that in scenarios where target domain training data is available, training on them is better, but in settings where that is not the case, the proposed method offers a good alternative wrt fairness, accuracy and interpretability.

# *Strengths.*

- The introduction was well written. The authors situated the problem well, their motivation was clear and well reasoned, especially in terms of importance. Language style supported ease of understanding.
- Discussion of related work appears very well organized, Table 1 is appreciated.


# *Weaknesses / Limitations.*

1) Given that the authors mention temporal diversity, I would appreciate seeing that being tested as well.  ( I personally don't know about the dataset in detail, going by how the authors described it)

   For example, the source model being trained on California domain at Time T1, targeted domains:  \-\> California T1 → California T2 (temporal shift, perhaps even at multiple Tx), and California T1 → Texas T2 (combined shift) would strengthen the claim of handling distributional shifts broadly.

   Does the method generalize similarly across temporal drift as it does across geographic drift?

# *Suggestions for Authors.*

Subjective presentation suggestions:

- For Tables, if the description could carry what you would like the reader to infer from the table as the last statement, it will make it easier to recognize the impact of your method, without it being spread through the discussions.

- Consider reorganizing results to highlight trends across sample sizes, perhaps combining Tables 2–4

- Highlight key takeaways more explicitly, maybe like, **\<Takeaway T1\>:** Talk about results in quantity and back up the claim.
  For example, instead of the subheading Few-shot adaptation (100 labeled samples), it may be more useful to say
  \*  **l1 regularization yields the best trade-off in few-shot adaptation (100 samples)**: {your justification based on results, walking over the table}

- The paper imo is well written in the first third,, but the last third does not do full justice to the authors’ contributions, as the main results are somewhat buried in text rather than highlighted as clear contributions.